# Urban Planning Policies to the Renewal of Riverfront Areas: The Lisbon Metropolis Case

**Eduardo Medeiros \***, **Ana Brandão**, **Paulo Tormenta Pinto and Sara Silva Lopes**

Centre for Socioeconomic and Territorial Studies, Instituto Universitário de Lisboa (ISCTE-IUL),
DINÂMIA'CET -Iscte, Edifício Sedas Nunes, Avenida das Forças Armadas, Sala 2W4-d,
1649-026 Lisbon, Portugal; ana.luisa.estevao@iscte-iul.pt (A.B.); Paulo.Tormenta@iscte-iul.pt (P.T.P.);
Sara_Alexandra_Lopes@iscte-iul.pt (S.S.L.)
**\*** Correspondence: eduardo.medeiros@iscte-iul.pt

**Abstract:** Urban planning offers various design possibilities to solve fundamental challenges faced in urban areas. These include the need to physically renew old industrial and harbour riverside areas into liveable, inclusive and sustainable living spaces. This paper investigates the way urban planning policies have helped to renew the waterfront areas in the Lisbon metropolis in the past decades. For this purpose, the contribution of the European Union (EU) and national urban development plans over the past decades are analysed. The results demonstrate an intense renewal of the waterfront areas in the Lisbon metropolitan area (LMA), particularly in Lisbon over the past three decades into leisure, ecologic and touristic areas, vis-à-vis the previous industrial and harbour vocation.

**Keywords:** urban planning; waterfront areas; Lisbon; urban development; POLIS; integrated sustainable urban development strategies

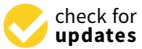



## 1. Introduction

Simply put, in almost every way, planning is a complex [1] and forward-thinking procedure, aiming to anticipate and tackle potential problems, envisioning the future [2]. In its generic meaning, planning is a set of interrelated processes [3] and a ubiquitous and highly political activity, aimed at improving the quality of decision making, with the ultimate goal of serving the public interest [4]. To be effective, spatial planning requires the integration of all territorial levels, as planning does not occur in pure isolation [5]. Furthermore, as a collaborative process, planning is profoundly preconditioned by various essential attributes such as the presence of a multi-disciplinary team, public participation, and data collection and analysis, among others [6]. In roughly equal parts, planning is directed to a plethora of targets, including "biodiversity, hazards/disasters, economic development, public health, water supply, energy, air pollution, climate change, . . . " [7] (p. 2, 27–28).

The need for a high level of city planning goes back several centuries [4]. Initially concerned with an orderly arrangement of public spaces, urban planning is now also focused on environmental imperatives, not only to create new environmentally friendly spaces [5], but also to anticipate potential environmental risks [8] with adaptive and resilient approaches [3]. Crucially, successful urban planning depends on how well it is adjusted to change, as well as on its flexibility and adaptability. Moreover, its success will depend on the prevailing planning culture since the enthusiasm for public planning varies from place to place [2]. In addition to cultural contexts, a complex network of socioeconomic, environmental, ethical, and behavioural urban imperatives affects the urban planning dilemma [3].

To be meaningful and feasible, urban planning must meet certain conditions. These include the capacity of urban spatial structures to exert some influence on aspects such as people's behaviour, the economy, well-being, and the natural environment. Secondly,

these structures should also be able to be influenced by external forces. Thirdly, urban planning processes require constant and effective monitoring and evaluation framework to assess their main impacts and finally, urban planning requires a multidisciplinary approach as it touches more than one scientific field [9]. Taking this further, some scholars advocate a gender perspective as a crucial element for urban planning [10].

The good news is that, by being an essential counterpart of urban development, urban planning processes have survived several financial crises, such as the one in 2008, which lead to some rethinking from urban planners [11]. It is, however, quite common to experience planning reforms every now and then, in part motivated by new political orientations or by the revival of political ideologies and orthodoxies [12,13]. In this regard, multifaceted critics of a rising neoliberal and growth-oriented planning paradigm have emerged over the past years and have raised alternative responses in view of more transformative urban planning visions [14]. In a different prism, since the mid-twentieth century, certain city planners were faced with an emerging mass tourism phenomenon, which required novel urban planning approaches to tackle its effects on dwellers as well as on the urban economy [15]. Additionally, noteworthy is the recognition of the inability of urban planning to address urban problems faced by many cities, such as housing, inequality, crime, and segregation. This calls for a normative reorientation of planning theory and practice [16].

Ultimately, "urban planning works to improve the welfare of people and their communities by creating places that are more convenient, equitable, healthful, efficient, and attractive for both present and future generations" [17] (p. 19). Suffice it to say that urban planning has a focus on space. In other words, urban planning is marked by physical planning processes [7]. These include urban conservation or regeneration processes towards, for example, sustainable housing [18], and even the more dramatic urban renovation processes [19]). Among these, are the construction or renovation of riverfront areas [20], as a means to support green mobility and ultimately improve the quality of life of city dwellers [21].

In its simpler understanding, riverfront planning encompasses policy actions aiming at managing the relationship between citizens and areas located close to a riverfront [22]. A cursory glance across existing literature on the renewal of riverfronts strengthens the argument for its crucial role in supporting sustainable urbanism. In particular, riverfront planning can provide increased quality of open urban spaces for pedestrians and cyclists [21], as well as the organisation of events such as concerts and festivals [20]. Indeed, recreational urban functions, such as public parks, sports facilities, playgrounds, are commonly located near riverfronts, thus attracting riverfront greenway projects that "have been able to also radically alter the character of the cities where they take place" [23] (p. 4). Besides the policy goals of reclaiming urban riverfronts for recreational use and public access, on certain occasions, urban planning actions have been taken to preserve the wildlife and habitats along the river, and also promote economic revitalization [24]. In essence, riverfront planning incorporates an interplay of economic, social and environmental aspects [25] which should incorporate five implementation planning principles: (i) accessibility; (ii) integration; (iii) partnership; (iv) participation and (v) construction [26].

The methodology of this article, drawing mostly on desk research (drawing on EU, national, regional, and urban planning policy documents and other literature, including studies and articles), assesses how far urban planning policies, particularly EU related initiatives, have contributed to the renewal of riverfront areas in the LMA, and more specifically in the city of Lisbon, since the early 1990s. The comparative case study methodology is adopted to investigate the experiences of urban planning in the renewal of waterfront areas in the LMA. In line with current urban planning literature approaches, the analytical framework accommodates a multitude of theoretical prisms involved in the implementation process and the examination of the results of the analysed urban planning processes. These include, among others, governance, participatory, environmental and financial aspects. The empirical analysis is supported by a comparative methodology of the analysed

urban policies (e.g., URBAN, POLIS, POLIS XXI, ISSUD). In addition, previously obtained knowledge on the paper's main subject by all authors was used along the written sections.

Following from this introductory note in which the concepts of planning and urban planning were introduced, as well as the methodological approach, the article is then divided into three main sections and a conclusion. The following section introduces and debates the main EU urban planning instruments which have been implemented in Portugal since the 1990s. The following two sections are dedicated to a more detailed analysis of the urban planning experiences in the renewal of waterfront areas, respectively in the LMA, and in the city of Lisbon. The two final sections discuss and summarise the main research conclusions.

## 2. EU Urban Planning Instruments towards a More Sustainable and Inclusive Lisbon

### 2.1. The EU Urban Policies in a Nutshell

Crucial development challenges to Europe, such as the need to tackle social segregation, poverty, environmental degradation, and improve resource efficiency and a green and circular economy, requires a strong urban focus, since more than 70% of Europeans dwell in urban areas, which generate two-thirds of the EU's GDP [27]. More importantly, however, are the advantages of an interconnected system of urban networks for the territorial development process in Europe, such as integrative geographic action platforms for new opportunities and creative solutions [28]. This is particularly important in Europe, due to its unique polycentric urban structure [29]. For some, however, the EU lacks a solid urban narrative [30]. Conversely, others purport a different perspective in which the urban question in the EU has attracted fundamental policy attention over past decades [31,32]. What is undeniable, however, is the constant EU support for urban development initiatives since the early 1990s, with a few highs and lows along the way [33]. One of the highlights being the publication of the Leipzig Charter, supporting the integrated urban development rationale [34]. This chart was recently updated as the New Leipzig Charter draft, reaffirming support for urban transformation through integrated urban development, with a place-based, multi-level and participatory approach [35].

Despite the relatively recent revival of the EU's growing interest in supporting urban development policies [36], animated by the launching of the Urban Agenda for the EU (May 2016) [37], the EU competencies in urban policies is limited. In fact, "there is no formal Council formation dedicated to urban policy and the engagement of different Member States has varied over time, the impact of the intergovernmental cooperation on EU and national policies have also been varied" [27] (p. 6). Even so, the European Commission (EC) has been supporting several policy initiatives aimed at financing urban development processes for some time. An eloquent example is the URBAN Community Initiative (1994–1999 + 2000–2007) [38], targeting neighbourhoods in extreme deprivation, with an integrated approach [39].

More recently, under the EU Cohesion Policy 2014–2020 period, the EC provided new momentum for supporting urban development, as it allocated approximately 10 billion euros from the ERDF to Integrated Strategies for Sustainable Urban Development—ISSUD, distributed among 750 European cities [40–42]. This integrated approach represents an opportunity to better encompass urban functional areas in managing urban development processes [43], and to ensure scale and focus [44]. More importantly, however, around half of the European Regional Development Fund (ERDF) is expected to be used in urban areas via the EU Cohesion Policy (2014–2020) mainstream operational programmes [45]. These urban targeted investments are complemented by other investments associated with an increasing number of EU sectoral policies [27]. Alongside, the EC has supported the previously mentioned Urban Agenda for the EU [46] establishing a novel multidimensional and multilevel cooperation platform for urban policy stakeholders [31], supported by the three pillars of EU policy-making: (i) better regulation; (ii) better funding and (iii) better knowledge [47]. From a territorial perspective, this agenda favours an integrated European urban policy [29]. A similar vision has made a mark on the EU URBACT programme that

has, for over 15 years, supported the sustainable integrated urban development rationale in cities across Europe in four main policy domains: (i) environment; (ii) governance; (iii) economy and (iv) inclusion [48].

### 2.2. The EU Urban Policies in the Lisbon Metropolitan Area

Portugal has been a formal member of the EU since 1986, benefiting from the EU Cohesion Policy funding since its first programming phase (1989–1993) and accessing other EU programmes aimed at territorial and urban development. In the LMA, this funding and initiatives have been used to support urban and environmental requalification operations, particularly on waterfront areas, as well as interventions in deprived areas (with some rehousing initiatives).

In the 1990s, Portugal accessed funding from 17 Community Initiatives (CI), including the URBAN CI, which targeted neighbourhoods in extreme socioeconomic deprivation in both the Lisbon and Porto Metropolitan Areas [40]. By joining the URBAN I (1994–1999) and II (2000–2006) rounds in the LMA, six deprived neighbourhoods were targeted, two of which were located within the Lisbon municipality (Table 1). For all political intents, existing evaluation reports concluded that, in overall terms, the URBAN CI contributed positively to socioeconomic integration, urban regeneration, urban greening, social qualification and the adoption of a previously absent integrated approach to urban development processes [49–51]. Not uncommon in similar EU programmes, the URBAN CI did not, however, solve all major socioeconomic problems affecting the targeted neighbourhoods [52].

**Table 1.** URBAN Community Initiatives (CI) Programmes in Lisbon Metropolitan Area.

| URBAN Programme | Neighbourhood | Total Investment (1000 Euros) |
|---|---|---|
| I | Lisbon—Casal Ventoso | 13,530 |
| I | Amadora—Damaia de Baixo | 3515 |
| I | Oeiras—Outurela/Portela | 19,165 |
| I | Loures—Odivelas | 5500 |
| II | Lisbon—Vale de Alcântara | 10,254 |
| II | Amadora—Damaia/Buraca | 5089 |

Source: Authors' compilation.

While the second URBAN CI was already underway, the POLIS programme was formally initiated in May 2000, with the main goal of supporting urban requalification to improve the urban environment in Portuguese cities [32]. Following a similar integrated policy approach as used in the URBAN programme, the POLIS programme was also implemented in two distinct phases and resulted in 40 interventions in 39 Portuguese cities [40], with an overall budget of €1.173 M [53]. Strictly speaking, the POLIS programme was vastly used to support an environmentally (greening) friendly urban planning approach, following the trends of EU policy rationale [54].

In turn, the subsequent POLIS XXI programme set a more coherent policy for cities [55] taking advantage of the funding from the 2007–2013 Cohesion Policy programming period. The programme embodied an intra-urban and city-region networking approach, promoting local and inter-municipal partnerships for the development of local strategies and action plans, involving different actors and agents. There were no shortages of POLIS interventions in the LMA. Indeed, the first programme targeted urban areas in the Sintra (including the Cacém urban area), Almada (Costa da Caparica), Setúbal, Moita, Barreiro and Vila Franca de Xira municipalities. Unlike the URBAN programme, however, the environmental character of the POLIS programme initiated urban greening renovation actions, including several waterside areas. In broad numbers, this programme contributed to: (i) requalifying 1,851,630 m$^2$ of public spaces; (ii) creating and improving 5,934,662 m$^2$ of green areas; (iii) creating 135,923 m of new walking paths; (iv) building bike lanes over an extension of 103,130 m; (v) requalifying 73,720 m of riverside areas; (vi) requalifying

15,850 m of seafront area; (vii) building new traffic conditioning areas totalling 150,170 m$^2$; and (viii) building 23,052 new parking spaces [56].

As for the POLIS XXI programme within the LMA, several redevelopment actions along the waterfront were financed through thematic line targeting areas of urban excellence (historic centres, riverfronts and seafronts, etc.), while two critical urban neighbourhoods (Cova da Moura—Amadora municipality and Vale da Amoreira—Moita municipality) received the large bulk of the total POLIS XXI investment, included in the "Iniciativa Bairros Críticos" [40].

The most recent (2014–2020) EU urban development programmes (ISSUD mentioned above), targeted 103 Portuguese municipalities, including all of the 18 Lisbon metropolitan municipalities. Seeking to encourage an integrated governance approach, coupled with the goal of strengthening capacities and collaboration among stakeholders at different territorial levels, the ISSUDs re-established the URBAN and POLIS goals of urban rehabilitation and the regeneration of disadvantaged urban areas in Portugal. Alongside, they reinforced monitoring/evaluation and a place-based urban policy implementation framework in all selected municipalities [41].

### 2.3. The EU Lisbon ISSUD

With regards to the Lisbon ISSUD (or PEDU—Plano Estratégico de Desenvolvimento Urbano—in Portuguese), its proposed strategy reflects previous urban planning instruments used in Lisbon which defined four main strategic priorities: (i) affirm Lisbon in global and national networks; (ii) regenerate the consolidated city; (iii) promote urban qualification; and (iv) stimulate participation and improve the governance model. These priorities translate into a wider vision to attract more people, making the city more attractive with improved quality of life, improving efficiency from an energy standpoint, and providing more protection against seismic risks. In this context, the Lisbon ISSUD strategy is supported by three main intervention axes [57]:

1. Attract more inhabitants: more accessible housing; parking for residents; a healthy urban environment and quality public facilities;
2. Capture more companies and jobs: promotion of business incubators and start-ups; support for spontaneous initiatives to reuse abandoned/vacant spaces;
3. Better city: boost urban rehabilitation; qualify public spaces in conjunction with mobility and urban regeneration; returning the riverfront to citizens; and promote sustainable mobility.

As with all the other Portuguese ISSUDs, the one for Lisbon included an Urban Regeneration Action Plan (PARU—Plano de Ação de Regeneração Urbana—in Portuguese) and an Integrated Action Plan for Disadvantaged Communities (PAICD—Plano de Acção Integrado para Comunidades Desfavorecidas—in Portuguese). The former (PARU), defined an intervention area corresponding to the central part of the Urban Rehabilitation Area (ARU) for Lisbon, delimited within the framework for the Lisbon Urban Rehabilitation Strategy (2011–2024), approved by the Municipal Assembly on 20 March 2012, through Resolution No. 11/AML/2012. The delimitations for this ARU were recently changed and published in the Official Gazette (Diário da República) No. 148, of 31 July 2015, second series (Figure 1).



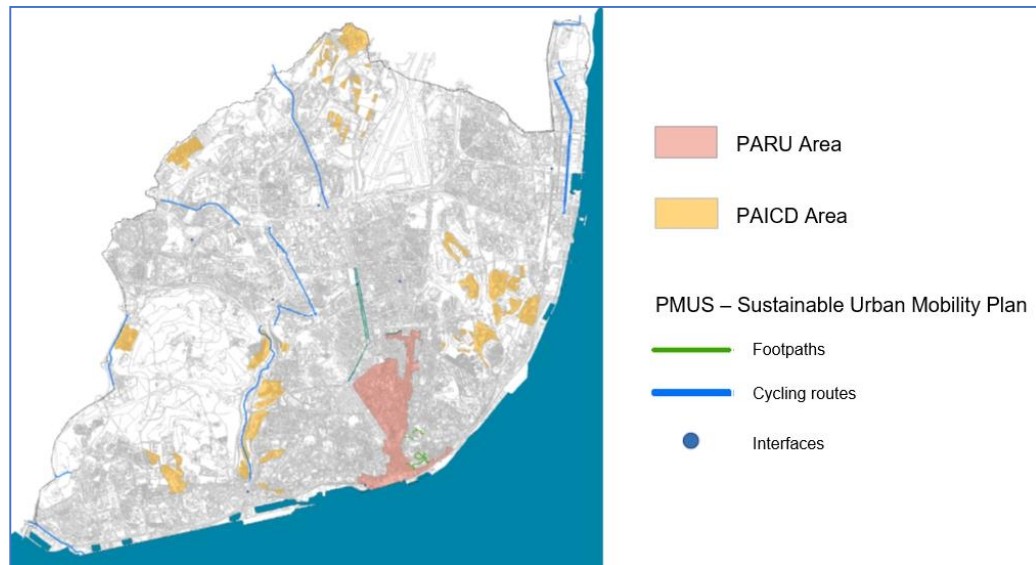

**Figure 1.** Lisbon PARU (Plano de Ação de Regeneração Urbana) intervention area within the Lisbon municipality. Source: [58].

The latter (PAICD), identified priority neighbourhoods and intervention areas within the city of Lisbon. Three intervention areas in three parishes within the city of Lisbon (Santa Clara, Marvila, and Santa Maria Maior) were selected for the PAICD (Figure 2).

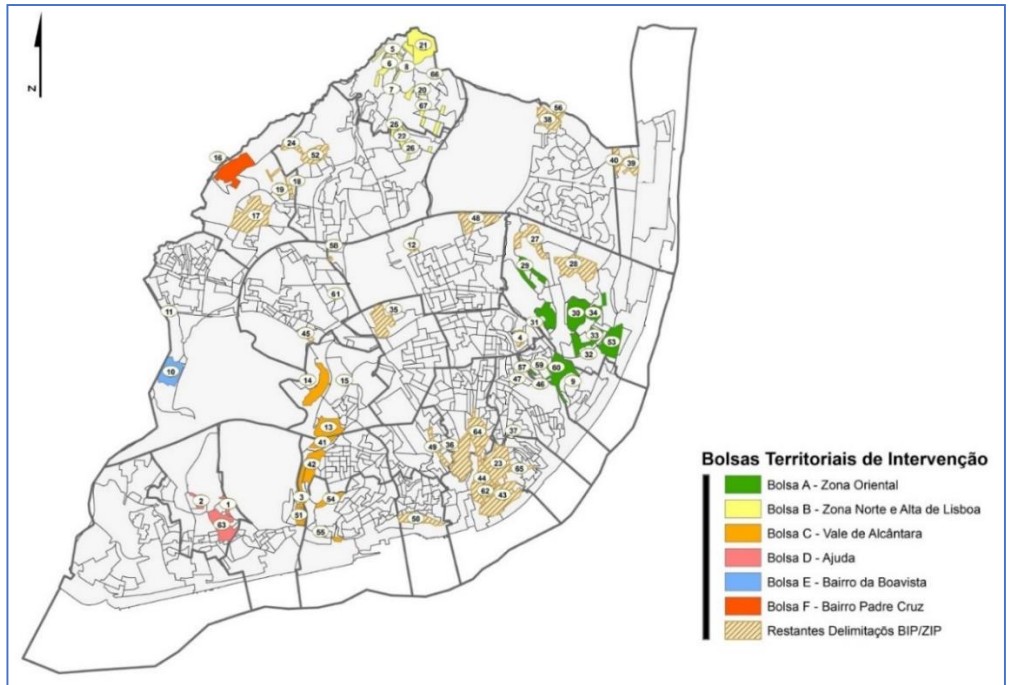

**Figure 2.** Lisbon PAICD (Plano de Acção Integrado para Comunidades Desfavorecidas) territorial pockets of intervention (Bolsas Territoriais de Intervenção in Portuguese) within the Lisbon municipality. Source: [59].

## 3. The Experiences of Urban Planning in the Renewal of Waterfront Areas in the Lisbon Metropolitan Area

The regeneration process of the riverfront area in the Lisbon metropolis must be considered within the overall post-industrial transformation, broadly addressed in the literature. The changes in sea routes, the evolution of logistic technologies, and competition

from aerial and road mobility forced a process of renovation upon harbour areas and their activities, freeing up zones and building stocks for new uses and new configurations [60]. The process began in the late 1950s in the United States, triggering a series of operations that, later on from the 1980s onwards, were extended to Europe, Asia and Latin America [61]. New interventions privileged the introduction of leisure spaces, culture and tourism, taking advantage of the water proximity as an important aesthetic and recreational feature [62–64]. Despite some common characteristics [65]—the importance of public space, the development of large-scale project-oriented operations, the strengthening of infrastructures (especially for mobility), strong image, etc.—different outcomes, processes and results are to be found [66].

The first example to fully embody this waterfront renovation spirit was the urban development project for the 1998 World Exposition in Lisbon (EXPO'98). The operation took place on the eastern riverside area, with the objective of revamping a 350 hectares industrial site into a multifunctional urban area, combining the exhibition venue with new post-event uses aiming to create a new urban centrality.

The exceptional nature of the operation called for extraordinary measures, with new planning instruments and urban management procedures. A master plan—Plano de Urbanização da Zona de Intervenção da Expo 98—defined the limits and areas of intervention to be developed with detailed specifications, according to which infrastructures, public spaces and buildings were developed. While only one of the plans corresponded to the exhibition site, and the others structured the surrounding urban areas, they all considered the feasibility of contemplating both the exhibition and its adaptation to the post-EXPO'98 period. To coordinate the entire intervention a public company was created, the Parque Expo SA, concentrating powers and competencies from central and local authorities within the operation area. Despite major investments in infrastructures and important environmental requalification work in the area, the design of the public space was the main element of the new urbanity with new typologies, establishing standards for urban design quality and waterfront renovation. Overall, the urban development project was considered by the government a success to be cherished and disseminated all over the country. Yet, some criticism arose for its inability to connect to the surrounding areas, the lack of diversity in the residential offer or the predominance of the financial interests of the private sector [67].

Within the LMA, the EXPO'98 example spurred several state and municipal initiatives for the renovation of waterfront areas, either alongside central and historic areas, or prompting the conversion of larger brownfield or natural devaluated spaces. The majority of these were financed mainly by EU funding and integrated into different programmes and policies.

In Almada, the Urban Renovation Programme (PRU Programa de Reabilitação Urbana) running from 1996 to 2000 an initiative similar to the URBAN actions financed by the European Economic Area Financial Mechanism (EEA Grants), prompted the integrated renovation of the historic center, bringing together economic and social actions with the physical renovation of the built environment. Along the riverfront, public and green spaces were created or redesigned.

However, it was the POLIS program (2000–2006) that financed significant waterfront renovation initiatives in several LMA municipalities. The program included different types of operations:

- major integrated operations managed by urban management firms—Costa da Caparica (Almada), Cacém (Sintra) and Setúbal;
- integrated operations managed by the municipalities—Vila Franca de Xira;
- local scale urban regeneration operations led by the municipalities—Barreiro and Moita.

The majority of the integrated operations include larger areas and more complex transformations and followed strategic and action plans prepared by a state company and local authorities. The plan stated the main goals of the operation: the urban plans to be prepared, the projects and actions to be pursued, and the financial and scheduling scheme. For the execution and management of the plan, urban management firms, known as

Sociedades Polis, were created combining participations from local government and the state (60% and 40% respectively). As for the integrated operations, they also included strategic and action plans, but the lower complexity of the operation was managed with a contract between the state and the municipality. Finally, local-scale operations were no different from ordinary city-led renovations which benefited from the program financing.

Set in different urban contexts, each operation had specific urban and environmental requalification goals, although shared characteristics may be found [68]. While major operations included the seaside and urban requalification of a coastal town (Costa da Caparica), urban restructuring and the environmental recovery of a dense suburban area (Cacém), the requalification of a historic core and the redesign of green areas (Setúbal), smaller operations were also carried out including waterfront promenades (Vila Franca de Xira), parks (Barreiro) and the redesign of public spaces (Moita). Creating connections to watercourses and green spaces, embodying a city-nature connection and the investment in public spaces and facilities (instead of new developments) are common to all cities.

The succeeding POLIS XXI (2007–2013) programme had an even greater impact on the transformation of waterfront areas in the LMA. This was due to a greater interest by municipalities to invest in these areas, but it was also connected to the definition of the policy and the financing mechanism, which within the scope of the Partnerships for Urban Regeneration included a specific entry for riverfronts and seafronts. Operations include the establishment of a local partnership and the definition of an action plan (including physical, environmental, economic and social dimensions) supported through a protocol with local, sectoral public and private actors. The selection of proposals was made through a public tender. In the LMA, ten municipalities successfully submitted 13 operations to this programme (Vila Franca de Xira, Lisboa, Alcochete, Montijo, Moita, Barreiro, Seixal, Almada, Sesimbra and Setúbal). These funded operations further encouraged the urban regeneration of riverside areas [69], investing in the improvement of the built environment in historic centres, the redesign of public spaces (new walkways alongside the water, larger pedestrian areas) with the inclusion of soft mobility features, urban greening actions but also the environmental recovery of natural habitats.

Overall, these operations and investments show the value and potential of waterfront areas for new uses, leveraging several brownfield redevelopment initiatives around the Tagus estuary. In 2008, an ambitious and large-scale operation was presented for three brownfield areas in the Tagus south bay (Quimiparque in Barreiro, Siderurgia Nacional in Seixal and Margueira in Almada). The Arco Ribeirinho Sul strategic plan was developed based on new major infrastructures: a rail and road crossing over the Tagus, the high-speed rail link and the Lisbon airport. Aimed at the urban development and economic dynamization of the metropolitan area, several urban planning schemes were developed forwarding a mix of uses—housing, commerce, services, equipment, and clean industries [70]. However, due to the effects of the financial and economic crisis, the major infrastructures were suspended changing the viability of these developments, forcing the revision of these initiatives.

Finally, the LMA ISSUD's were developed through the Urban Development Strategic Plans (or PEDU—Plano Estratégico de Desenvolvimento Urbano—in Portuguese) in which each municipality frame the urban regeneration and the strategies for the improvement of deprived urban areas, also entailing the basis for the application for EU funds under the Portugal 2020 Programme. The PEDU includes the Urban Regeneration Action Plan (PARU—Plano de Ação de Regeneração Urbana—in Portuguese), the Integrated Action Plan for Disadvantaged Communities (PAICD—Plano de Acção Integrado para Comunidades Desfavorecidas—in Portuguese) and the Sustainable Urban Mobility Action Plan (PAMUS—Plano de Ação de Mobilidade Urbana Sustentável—in Portuguese) (Table 2 and Figure 3).

**Table 2.** EU urban policies and urban planning initiatives in Lisbon Metropolitan Area.

| Programme | Municipality | Operation |
|---|---|---|
| PRU | Almada | Nova Almada Velha (01) |
| POLIS | Almada | Polis da Costa da Caparica (02) |
| | Sintra | Polis do Cacém (03) |
| | Setúbal | Polis de Setúbal (04) |
| | Vila Franca de Xira | Polis Vila Franca de Xira (05) |
| | Barreiro | Polis Barreiro (06) |
| | Moita | Polis Moita (07) |
| POLIS XXI | Vila Franca de Xira | Requalificação da Frente Ribeirinha da Cidade de Vila Franca de Xira (08) |
| | | Requalificação da Frente Ribeirinha da Zona Sul do Concelho (09) |
| | Lisboa | Operação Integrada Ribeira das Naus/Terreiro do Paço (10) |
| | Almada | Revitalização de Almada Velha—Ginjal: Cultura Lazer Turismo (11) |
| | Seixal | Acção integrada da Regeneração e Valorização da Frente Ribeirinha Seixal-Arrentela (12) |
| | | Regeneração Urbana—Valorização da Frente Ribeirinha de Amora (13) |
| | Barreiro | REPARA—Regeneração Programada da Área Ribeirinha de Alburrica (14) |
| | Moita | Operação de Valorização Integrada da Zona Ribeirinha: Da Caldeira da Moita à Praia do Rosário (15) |
| | Montijo | Requalificação e Dinamização da Frente Ribeirinha do Montijo (16) |
| | Alcochete | Acção integrada de Regeneração Urbana—Valorização da Frente Ribeirinha de Alcochete" (17) |
| | Setúbal | Programa de Regeneração Urbana do Centro Histórico de Setúbal (18) |
| | | Programa Integrado de Valorização da Zona Ribeirinha de Setúbal (19) |
| | Sesimbra | Programa Integrado de Valorização da Frente Marítima de Sesimbra (20) |
| ISSUD | Alcochete | PARU (21) |
| | | PAICD (22) |
| | Almada | PAICD (23) |
| | | PARU (24) |
| | Barreiro | PARU (25) |
| | | PAICD (26) |
| | Cascais | PARU (27) |
| | Lisboa | PARU (28) |
| | Loures | PARU (29) |
| | Mafra | PARU (30) |
| | Moita | PAICD (31) |
| | | PARU (32) |
| | Montijo | PARU (33) |
| | | PAICD (34) |
| | Seixal | PARU (35) |
| | | PAICD (36) |
| | Oeiras | PARU (37) |
| | Sesimbra | PARU (38) |
| | Setúbal | PAICD (39) |
| | Vila Franca de Xira | PARU (40) |

Source: Authors' compilation.

Each PARU identifies areas, previously delimited as ARU's (or parts of it), where an integrated upgrade to the quality of the urban environment is intended, and specifies actions for that improvement. Once an important part of the urban nucleus of the LMA in the proximity of the riverfront, several PARU's foresee actions for these areas. Aligned with the urban regeneration and sustainable development targets, initiatives provide support for the renovation of buildings, local facilities, improvement to the proximity of public space and the promotion of sustainable mobility.

As for the PAICD, it identifies disadvantaged neighbourhoods or local communities specifying responses within the physical, economic, and social dimensions. As the location of these areas in proximity to the waterfront is not so common (especially with regard to public housing estates, normally situated in peripheral areas), fewer municipalities

can be identified. In these cases, they include the improvement of the built environment, buildings renovation and local facilities in addition to other initiatives of social nature.

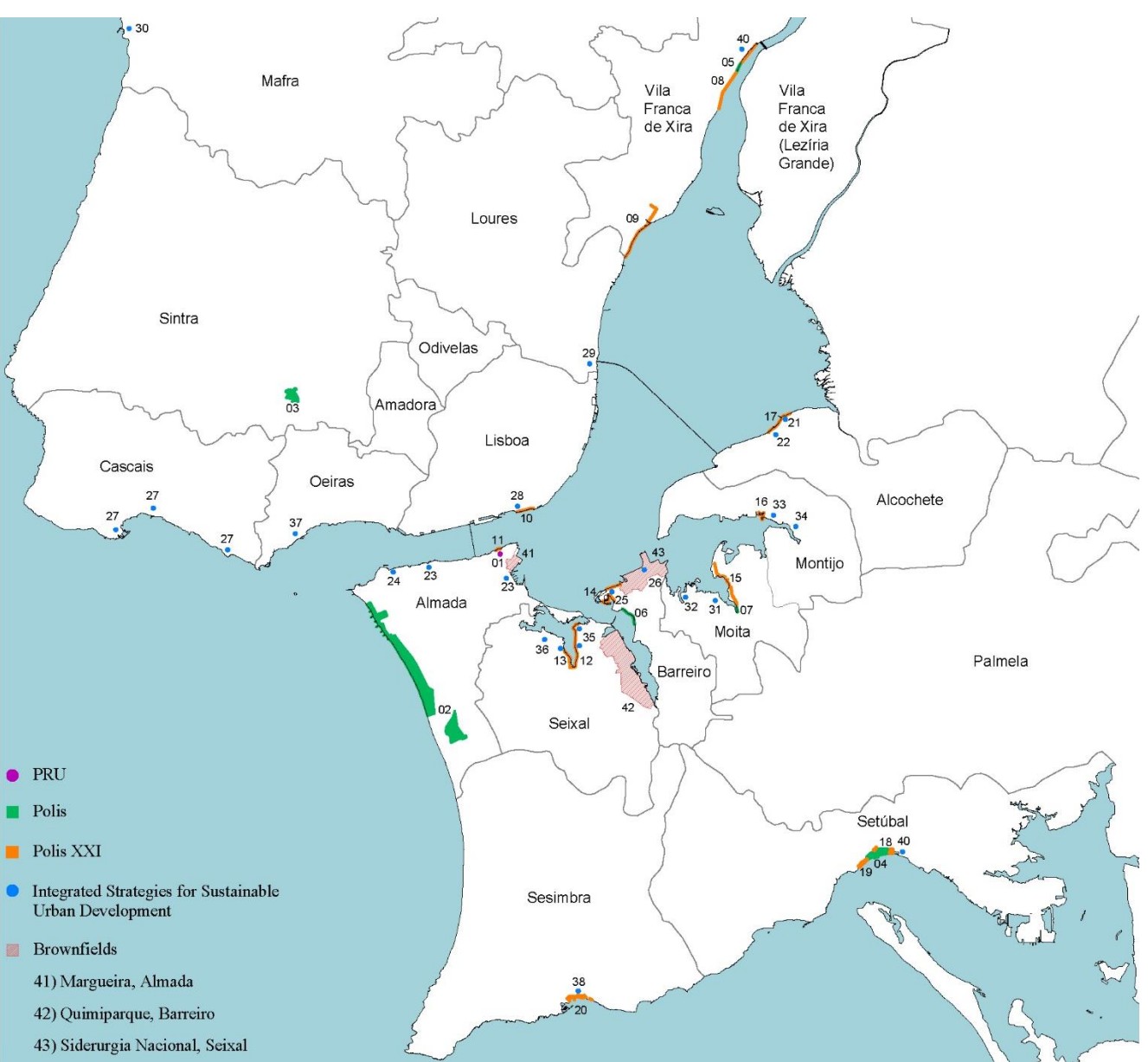

**Figure 3.** Urban planning programs in the renewal of waterfront areas in Lisbon Metropolitan Area. Source: Authors. Note: the numbers on the figure represent specific waterfront renewal programs, which are identified in Table 2.

## 4. The Renewal of Waterfront Areas in Lisbon

Investments made in the last decades in urban regeneration have brought important changes to Lisbon's waterfront spaces and improvement to the built environment, enhancing the city's image and competitiveness. These transformations are common to many other urban areas, as is evident in a growing body of literature, supported by city strategies for improving the attractiveness of cities: organization of mega-events, the redevelopment of large brownfields, the development of new cultural and sports venues, the construction of new infrastructures, with favourable consequences in the growth of the real estate and tourism sectors and the strengthening of urban entrepreneurial culture.

In the case of Lisbon, the 1990s were the turning point in these policies prompting discussions on the reconversion of harbour sites also connected to the development of a set

of new planning instruments. A first example is the 1988 competition for the renovation of the Lisbon riverfront organized by the Portuguese Architects Association. Soon after, the Strategic Plan (1992) and the Municipal Master Plan (PDM, Plano Director Municipal—in Portuguese) (1994) presented the riverfront as a fundamental urban area for planning and structuring, identifying the need to reinforce city–port connections [67]. Together with these plans, regeneration and place-branding efforts were reinforced with the hosting of major events, Lisbon 1994, European Capital of Culture, EXPO'98, and later the 2004 UEFA European Championship.

As stated, the EXPO'98 flagship project set the example for the transformation of Lisbon's waterfront, particularly regarding the new public and green spaces close to the river and the mixed-use urban development. However, expanding this urban design to the whole of the 20 km of the city's waterfront, breaking down infrastructural barriers and transforming strictly harbour areas into spaces with a greater diversity of uses took several years to be developed. The inexistence of specific and detailed planning instruments, the jurisdiction of the Lisbon Port Authority and the lack of agreement between the different actors on the vision for the development of these spaces, were part of the problem.

These institutional and practical constraints are overcome with the development of the 2008 General Plan for the Lisbon Riverfront (PGIFR—Plano Geral de Intervenções da Frente Ribeirinha de Lisboa—in Portuguese), a comprehensive plan entailing the entire riverfront of the city and the urban fabric in the vicinity. The document [71] reflects the agreement and protocol between the Lisbon City Council and the Lisbon Port Authority for the management of the city's riverfront, marking areas reserved for port activities (under port jurisdiction) and other areas for reconversion into mixed-use and public enjoyment (transferred to municipal responsibility) while signalling on-going and future urban development projects. Although not legally binding, the plan strengths the valorisation of the environmental, cultural heritage and the landscape along the riverfront as one of the strategic projects for the municipality [72] (Figure 4).

For each of the main riverfront areas, the plan identifies the problems to be addressed; the goals of the transformation and the specific proposals or intentions to be elaborated; specifically detailed urban plans and schemes; projects for urban design and public space and large public facilities and private developments.

Regarding the urban planning instruments, several master plans and detailed plans cover the important areas of the river–city interface. While in most of the historic and central areas, the plans focus on heritage conservation and urban regeneration, in other areas, they foster urban restructuring and small-scale redevelopment or even large-scale development of former industrial areas.

Municipal action was particularly strong and sustained with regard to the investment and requalification of public spaces, starting with the historical centre and then spreading to other areas of the city. Most of the projects aim to reduce car movement and increase spaces for pedestrians and bicycles, as well as green spaces and urban greenery, and improve commercial and leisure activities or their integration with green mobility structures.

This includes the riverfront projects funded by the POLIS XXI program and the ISSUD strategy for Lisbon. In the first case, the Integrated Operation between Ribeira das Naus and Terreiro do Paço (Figure 5a) was part of the central and monumental riverside area renovation, aiming to introduce new uses and activities and reconnecting the city physically, functionally and visually with the river. The public space project, from PROAP and Global landscape architects, designs a green space and waterfront with a gentle slope towards the river, creating rest and leisure points in the proximity of Lisbon's main square Terreiro do Paço. The memory of the place (a former shipyard) is invoked through the exhibit and integration of archaeological structures. In the second, the Public Space Regeneration Operation—Cais Sodré | Corpo Santo (Figure 5b) extends the prior redesigned area to one of the most important transportation hubs in the city (Cais Sodré). In line with the regeneration strategy, the project from Bruno Soares Architects, redesigns vehicular circulation to provide more comfortable pedestrian movement and connections to public

transport and soft mobility modes, while providing new riverfront spaces and green areas for social enjoyment and rest.

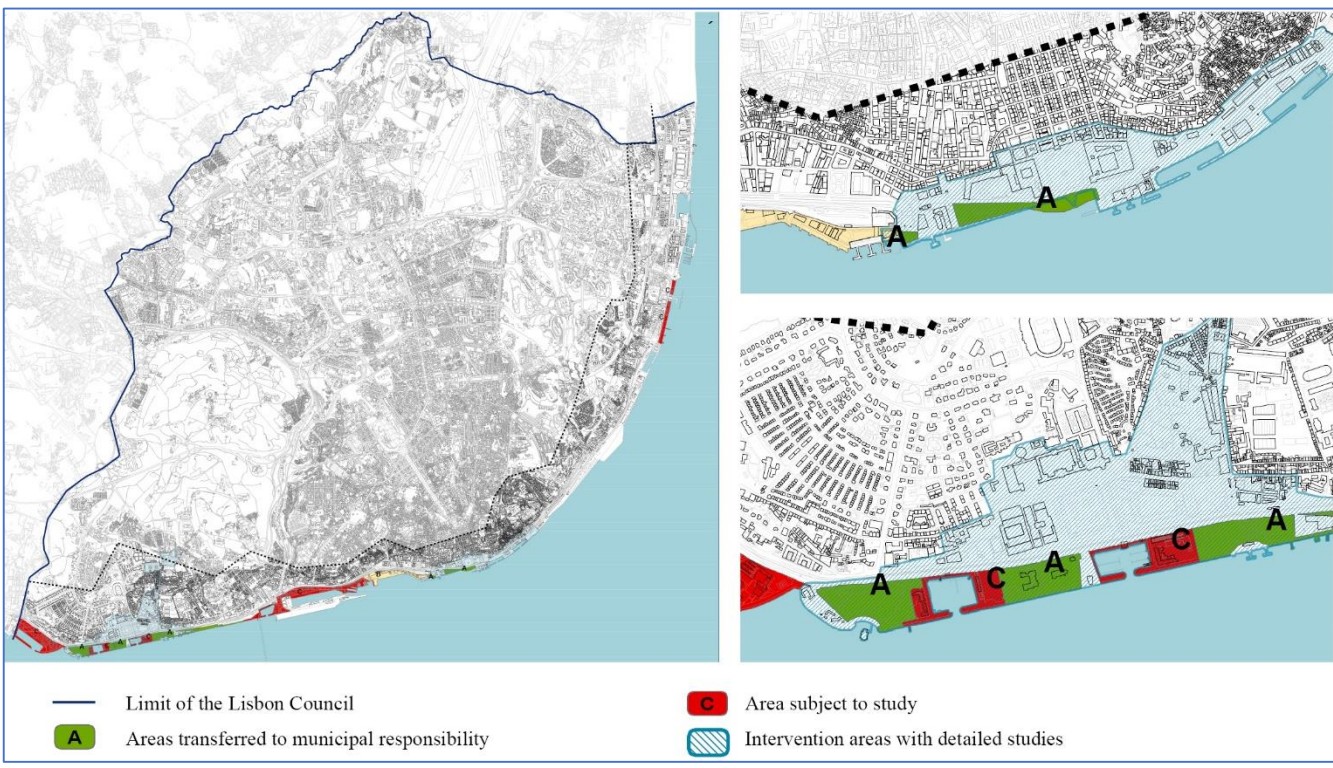

**Figure 4.** General Plan for the Lisbon Riverfront (PGIFRL). Areas under municipal management and port jurisdiction. Source: Adapted from [71].

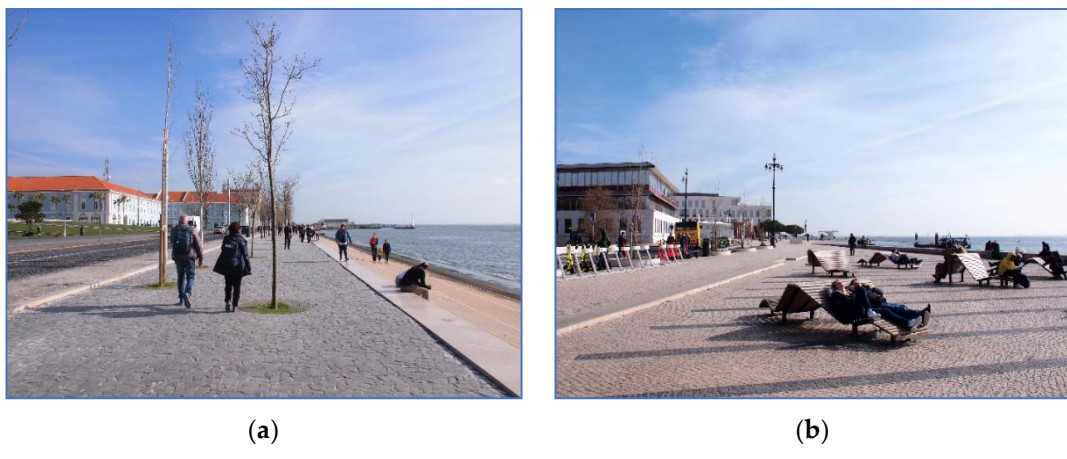

(**a**)                              (**b**)

**Figure 5.** Ribeira das Naus waterfront promenade (**a**) and Cais do Sodré public space (**b**). Image credits: Authors.

The changes are not limited to the urban structures and the public space of the city, several new buildings, works of considerable size, have been changing the city's riverfront structure, uses and image. For instance, the investments in transport and mobility infrastructures (intermodal stations, underground stations, elevators) improved movements in the city and even in the LMA and facilitated pedestrian and cycling enjoyment of this area; while cultural buildings, mostly new museums added to the existent facilities in increasing leisure and tourism uses. Corporate buildings and other privately owned facilities—a research centre and a hospital—showcase that private actors also value the riverfront location. Driven by this valorisation, together with other investment and urban

renewal policies [73], the existing building stock is also subject to change, although in a more piecemeal renovation approach, with increasing commercial, housing (high income and luxury, short-term rentals) and touristic uses. Finally, a number of large-scale urban developments are set to take place, converting disused industrial and port areas and other urban voids: private operations target high segment housing areas while public-led initiatives focus on mixed-use areas or environmental requalification.

The combination of these investments—public and private—reveals that the waterfront has become a more dynamic and highly visible territory in the city of Lisbon, combining areas dedicated to tourism, recreational and leisure activities, others undergoing a strong change of use, while maintaining port activities (Figure 6). Many of these buildings were designed by renowned Portuguese and international architects (e.g., Charles Correa, Paulo Mendes da Rocha, Amanda Levete, Renzo Piano, Aires Mateus or Carrilho da Graça) signalling its distinctiveness. Therefore, the role of architecture was decisive in this transformation, exploring new shapes and typologies, with a renovated sense of monumentality and iconographic status, often with recognizable authorship. This attractiveness and its connections to different public policies on regeneration and place-branding, and to private agent actions should be further taken into account, regarding the positive and negative effects on the city's (and metropolitan area) sustainable development and social cohesion.

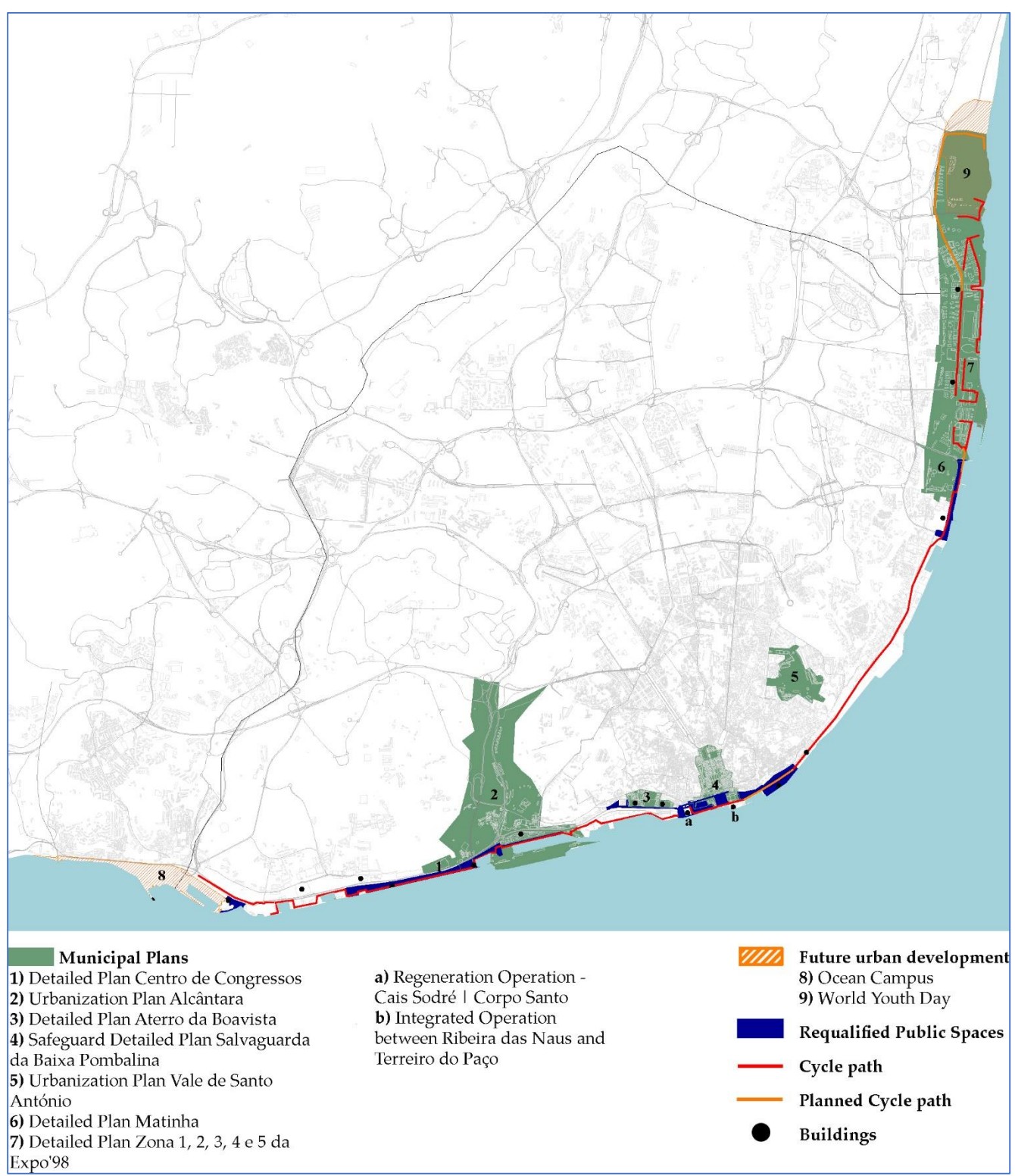

**Figure 6.** The renewal of waterfront areas in Lisbon. Source: Authors.

## 5. Discussion

Our survey of the waterfront urban planning process and urban change of the city of Lisbon and its associated Metropolitan Area led us to understand the profound functional transformations of these areas and their linkage to different scopes of planning policies and urban design practices. While most initiatives can be viewed as successful urban regeneration processes, they also showcase the inherent complexities and conflicts of

planning and executing highly contested urban areas [74] such as these. Therefore, in this section, we discuss how the Lisbon case embodies different topics, challenges and main trends debated over the last decades on waterfronts redevelopments.

The EXPO'98 case is representative of the late 20th century urban policies [75], taking the organization a major international event to launch a large flagship development project [76] with an urban entrepreneurialism culture [77], redesigning the urban waterfront with a high-quality built environment with new economic, social, aesthetic and symbolic values. The multifunctional area became a new centrality, mainly at the metropolitan level, mostly due to the well-designed and qualified public spaces and urban facilities [78], attracting a population far beyond that neighbourhood.

Nevertheless, this urban regeneration process also had disadvantages. At the residential level, it has become a homogeneously middle- and upper-class income area, benefiting from a scenario of quality and exclusivity and the lack of social and physical connection to the surrounding urban fabric. Although the neighbourhood does not correspond to a gated community (nor to privatized public spaces), Gato [79] identifies a process of social selectivity in line with other examples of appropriation of renewed waterfronts by groups with higher economic power [80]. Another unattained goal was the incapacity of the operation to promote the regeneration of the Eastern area of Lisbon (on the riverfront and to the segregated social housing areas), as intended [81]. Like a typical flash-ship project, the EXPO'98 operation concentrated much of the resources and opportunities that could have been distributed to other areas.

Regardless of these more negative appreciations, the tangible and intangible legacy of the EXPO'98 operation is considered as generally positive, namely regarding the innovations and diffusion of a culture based on the quality of the built environment. In fact, many features and planning and design procedures were considered to be exemplary, setting a standard to many operations in the Lisbon area and all-around Portugal [68]. In particular, the successful waterfront public spaces of this area were set as role models, leveraging the process of rehabilitation of open spaces in the riverfront of Lisbon.

The following decades Lisbon waterfront change—in its political, aesthetic and economical issues—is somehow linked to this initiative to gain prestige and relocate the city on the map of competitive globalization, in what Moulaert, Rodríguez and Swyngedouw [75] describe as global-local restructuring processes and it is part of a wider set of urban planning and investment policies, with effects on tourism flows, economic activities and the real estate market.

While the physical transformation required the progressive deactivation of port areas (some already with no activities) and its allocation to new uses, it also entailed negotiation on land ownership, future visions and interests by the different agents of the area, mainly public (central and local government, port authorities), showing the complex set of power relations between stakeholders [82]. Several authors signal the change of perspective between the waterfront as a traditional space of work versus a new space for leisure and consumption, perceived as "new lifestyle centers" [83]. In this view, the presence of water is mostly valued by its scenic quality [84] and aestheticized alongside iconic buildings or redesigned open spaces while the riverfront becomes a central element of the urban landscape.

Albeit the commodification of the Lisbon waterfront is a part of the contemporary debate [85], the ongoing transformation combines public access and enjoyment to qualified riverside areas, star-architect projects as symbols of this new image [86], redevelopment projects and undefined areas. The balance between the mix of uses, and the public for these uses, needs to be clearly defined [87], as well as who benefits from these transformations, private or public interests [83] and who gets to participate in the planning and design process [88].

As for the process of waterfront redevelopment in the LMA, despite the connections to the main city, it presents particular challenges from the ones debated above. On one hand, the legacy of the EXPO'98 operation and other success stories of waterfronts regeneration

processes [89], fosters different municipalities to engage in the transformation of these areas. However, in these cases, mainly through local scale, public space and facilities projects, targeting urban and environmental regeneration of waterside areas more or less in proximity to urbanized areas. Therefore, most of these actions do not correspond to the typical large-scale urban development, nor directly engage real-estate and built environment upscale. On the contrary, the large brownfields redevelopment operations launched, were not able to guarantee the means and resources for the pursuit of those planning schemes.

On the other hand, many of these new spaces are linked to leisure, recreation, culture, and some commercial activities (mostly restaurants); in recent years, tourism has gained relevance, also as a spill-over phenomenon from Lisbon's touristic boom. Despite being highly used areas by local populations, the projects have not created conditions for more diversified activities, nor engage with more productive uses [90]. Yet, despite the predominance of this kind of activities as well as types of spaces—promenades, parks, cycle and footpaths—the universal image of the waterfront, commonly reproduced in many cases [91], is fairly balanced out with local identity features and mostly natural assets.

From a social justice perspective, these redesigned waterfronts present opportunities to assure wide access to quality public spaces and enjoy natural amenities, taming inequalities created by poorly planned urban development and highly industrialized areas [92]. As this balance between the ecological role of urban waterfronts and its human activities is becoming more evident by the climate change debate, Pinto and Kondolf [87] stress the importance to improve the ecological functions of these areas by combining desirable urban activities and infrastructure, with watercourses preservation or restoration actions.

Overall, the Lisbon city and metropolitan area's waterfront urban change showcases that nor the processes of transformation and redevelopment are the same within the different areas, nor the outcomes of those projects are unanimous. In line with other authors [74,84,93], our analysis supports the ongoing debate about the multifaceted nature of waterfronts, encompassing physical, economic, social, political, environmental topics. As the sustainability of these interventions is becoming more and more central on the contemporary debate, more contradictions and conflicts appear evident, much remains to be managed by urban planning policies in order to address the complexity of urban waterfront developments.

## 6. Conclusions

This article presents a comprehensive overview of the urban planning policies, on all territorial levels, that have contributed to the renewal of the waterfront areas in the LMA and the city of Lisbon, in particular. It initiates with the analysis of the strategic vision, implementation process, and main effects of EU and national urban development policies since the early 1990s, such as the URBAN CI, POLIS, POLIS XXI and the ISSUDs, in the LMA. The conclusions are that they all added an integrated policy approach to urban development, and contributed to physically rehabilitating several urban areas, predominantly improving the environmental aspects of certain urban neighbourhoods. Conversely, and despite some positive indicators, the interventions made to the deprived neighbourhoods did not trigger substantial changes in socioeconomic conditions.

By the same token, the central finding for the EU urban policy planning interventions is their positive contribution to physically rehabilitate certain urban areas with an integrated policy approach, by financing previous urban development strategies. This was particularly evident when analysing the LMA, where the continuous framework and financing support from the EU policies had a direct impact on the transformation of metropolitan waterfronts, combining new recreational and leisure uses, with the environmental recovery of urban and natural spaces, projecting a new image of the metropolitan territories. In some cities, several partial projects executed successively, ultimately design continuous structures and large-scale waterfront spaces. An example is the riverside promenades and parks which link almost all of the entire riverside area of the northern bank of the Tagus estuary.

This is also the case for the city of Lisbon, which has used the current ISSUDs, not so much to follow novel and innovative policy approaches for urban development, but to consolidate established urban development strategies towards a more attractive international city for inhabitants and businesses, via the (re)qualification of public spaces and improved and sustainable mobility. Here, one idea that has animated this urban modernisation process is giving back the riverfront to citizens, in a city dominated by a large and historic river estuary on its southern and eastern borders. However, despite being the municipality's strategy to "give the river back to the city", the implementation of this transformation also involves the action of other public entities (port authorities, tourism agencies, transport companies) and several private agents (large companies, private institutions and individual players).

Therefore, the development of waterfronts has become a primary source for driving urban transformations, viewed by public and private agents as a place branding opportunity and a site for visibility of novel urban policies while delivering an innovative image of the city. However, the development of sustainable urban policies and the compatibility between the interests of the different actors and agents is not straightforward, entailing further efforts and better urban planning mechanisms to address contradictory views and unintended negative impacts.

In this view, at least three suggestions for future research can be identified from this study. One deals directly with the knowledge gap related to how public and private agents can effectively cooperate to deliver more sustainable and integrated urban plans related to the riverfront renovation, while avoiding setbacks and inadvertent negative impacts. A second discussion topic for future research is the need to explore urban planning participatory approaches encompassing citizens' views in an effective manner. Thirdly, metropolitan areas dominated by an estuary, like the case of Lisbon, require further research on how to effectively pursue various interests concerning social, economic and environmental aspects related to the implementation of riverfront renovation with multi-level governance and integrated policy approach.

**Author Contributions:** Conceptualization, E.M. and A.B.; methodology, E.M. and A.B.; validation, P.T.P., A.B. and S.S.L.; formal analysis, E.M. and A.B.; investigation, E.M. and A.B.; writing—original draft preparation, E.M.; writing—review and editing, A.B. and S.S.L.; figure realization—A.B. and S.S.L.; visualization, E.M. and A.B.; supervision, P.T.P. and S.S.L. project administration, E.M. All authors have read and agreed to the published version of the manuscript.

**Funding:** The authors would like to thank the anonymous reviewers for their careful reading of the manuscript and for their valuable comments and suggestions. This research was funded by Grant PTDC/ART/DAQ/32561/2017 by National funds through the Portuguese Science Foundation and Technology (FCT).

**Institutional Review Board Statement:** Not applicable.

**Informed Consent Statement:** Not applicable.

**Data Availability Statement:** Not applicable.

**Conflicts of Interest:** The authors declare no conflict of interest.

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
