# Peer review of "Urban Planning Policies to the Renewal of Riverfront Areas: The Lisbon Metropolis Case"

_sustainability, doi:10.3390/su13105665_

Round 1

Reviewer 1 Report

Thank you for the opportunity to read your manuscript.  Although it is descriptive, it is very focused on the Lisbon case and should lend some planning implications in general which should be improved upon.  

You could strengthen the conclusions by discussing some of the contradicting views and unintended consequences.  For example, the Expo98 site went through much displacement and gentrification.  How has the city dealt with that and what has been the positive and negative impacts 20 years later?

Author Response

Reviewer 1
- You could strengthen the conclusions by discussing some of the contradicting views and unintended consequences. For example, the Expo98 site went through much displacement and gentrification. How has the city dealt with that and what has been the positive and negative impacts 20 years later?
R: A discussion section was added to debate the complexity of the waterfront planning and redevelopment, including a section about the legacy of the Expo’98. Additionally, in section 3 more details inform about the criticism on the operation.

Reviewer 2 Report

Here are some minor revisions:

  • in section Introduction, I suggest including some references for the issue of renewal of riverfront areas, and describe this theme better.
  • Legend and numbers in Figure 6 need a better quality.
  • Section References needs an editing review.
  • Conclusions should define the future of the research. Please, complete this aspect.

Author Response

Reviewer 2
- In section Introduction, I suggest including some references for the issue of renewal of riverfront areas, and describe this theme better.
R: An additional paragraph was added in the introduction on the issue of renewal of riverfront areas with additional references
- Legend and numbers in Figure 6 need a better quality.
R: The figure was rearranged accordingly
• Section References needs an editing review.
R: References will have to be changed into Sustainability guidelines if the paper is accepted for publication and further editing will take place there
• Conclusions should define the future of the research. Please, complete this aspect.
R – An additional paragraph was added to the conclusion with comments on the future of the research

Reviewer 3 Report

Dear Authors,

I read the article with interest. It is interesting.

At this point, however, I would turn to issues that require improvement.
I believe that in Chapter 2 (and Introduction) the topic of river/waterfront planning could be extended to include international literature. Of course, I find such references, mainly in the Introduction, but they are insufficient. This subject has developed a huge scientific output, explanations and discourses. It is worth extending it.

The second point is the lack of Discussion. On the basis of the manuscript, I conclude that the Authors have extensive knowledge and experience in the presented topic. For Sustainability readers, it would be valuable to get to know the opinion and attitude of the Authors to the given issues, but in relation to international scientific achievements.
The conclusions are correct but, in my opinion, insufficient. Perhaps a discussion with the findings of the researchers who published their articles and books on waterfronts could even enrich the Conclusions.

The third issue is the lack of a separation part or at least a broader description of the research method. It is necessary for a scientific article.

I appreciate valuable figures. In Fig. 3 there is no explanation of what the reference numbers mean.

Authors are encouraged to make appropriate improvements to the manuscript.

Author Response

Reviewer 3 - I believe that in Chapter 2 (and Introduction) the topic of river/waterfront planning could be extended to include international literature. Of course, I find such references, mainly in the Introduction, but they are insufficient. This subject has developed a huge scientific output, explanations and discourses. It is worth extending it.
R: An additional paragraph was added in the introduction on the issue of renewal of riverfront areas with additional references. References were also added in the beginning of section 3. The international literature is more thoroughly debated in the new section 4 discussion. - The second point is the lack of Discussion. On the basis of the manuscript, I conclude that the Authors have extensive knowledge and experience in the presented topic. For Sustainability readers, it would be valuable to get to know the opinion and attitude of the Authors to the given issues, but in relation to international scientific achievements. The conclusions are correct but, in my opinion, insufficient. Perhaps a discussion with the findings of the researchers who published their articles and books on waterfronts could even enrich the Conclusions.
R: A discussion section was added to include a debate on how the Lisbon waterfront case connects and relates to the national and international research. - The third issue is the lack of a separation part or at least a broader description of the research method. It is necessary for a scientific article.
R: A few lines were added in the paragraph explaining the research method which already covers the means used for this research.
“Methodology draw mostly on desk research (drawing on EU, national, regional, and urban planning policy documents and other literature, including studies and articles), this article assesses how far urban planning policies, particularly EU
related initiatives, have contributed to the renewal of riverfront areas in the LMA, and more specifically in the city of Lisbon, since the early 1990s. Comparative case study methodology is adopted to investigate the experiences of urban planning in the renewal of waterfront areas in the LMA. In line with current urban planning literature approaches, the analytical framework accommodates a multitude of theoretical prisms involved in the implementation process and the examination of the results of the analysed urban planning processes. These include, among others, governance, participatory, environ-mental and financial aspects. The empirical analysis is supported by a comparative methodology of the analysed urban policies (e.g. URBAN, POLIS, POLIS XXI, ISSUD). In addition, previously obtained knowledge on the paper main subject by all authors was used along the written sections. - I appreciate valuable figures. In Fig. 3 there is no explanation of what the reference numbers mean. R: A note was added on Figure 3 to explain what the numbers are.

Round 2

Reviewer 3 Report

Dear Authors,

I consider the amended text to be significantly improved. I have no more comments.